🔓 | **Open Peer Review** | Ecology | Research Article

# Investigation of the genetic diversity of gut mycobiota of the wild and laboratory mice

Ji-Xin Zhao,[1] Hany M. Elsheikha,[2] Kai-Meng Shang,[1] Jin-Wen Su,[1] Yong-Jie Wei,[1] Ya Qin,[1,3] Zi-Yu Zhao,[3] He Ma,[1] Xiao-Xuan Zhang[1]

**ABSTRACT**    Mice are colonized by diverse gut fungi, known as the mycobiota, which have received much less attention than bacterial microbiota. Here, we studied the diversities and structures of cecal fungal communities in wild (*Lasiopodomys brandtii*, *Apodemus agrarius,* and *Microtus fortis*) vs laboratory C57BL/6J mice to disentangle the contributions of gut fungi to the adaptation of mice to genetic diversity. Using ITS1 gene sequencing, we obtained 2,912 amplicon sequence variants (ASVs) and characterized the composition and diversity of cecal mycobiota in mice. There were significant differences in the composition of cecal fungal communities between wild and C57BL/6J mice, with more species diversity and richness of fungi in wild mice than C57BL/6J mice. We cultured 428 fungal strains from the cecal mycobiota, sequenced the whole genome of 48 selected strains, and identified 500,849 genes. Functional annotation analysis revealed multiple pathways related to energy metabolism, carbohydrate metabolism, fatty acid metabolism, and enzymes involved in the degradation of polysaccharides, lipids, and proteins, and secondary metabolite biosynthesis. The functions and abundance of *Hypocreales* and *Pleosporales*, which included the majority of the crucial metabolic pathways, were significantly higher in wild mice than in C57BL/6J mice. The results suggest that variations in the fungal community composition may relate to the adaptability of mice to their environmental habitats.

**IMPORTANCE**  In this study, we analyzed the fungal microbiota of three wild mouse species alongside laboratory mice using ITS1 amplicon sequencing. By integrating whole-genome sequencing with culturomics, we sequenced the genomes of 48 fungi isolated from cultured strains and investigated their biological functions to understand the role of intestinal fungi in the environmental adaptability of wild mice. This investigation has expanded the functional gene repository of gut fungi and shed new light on the intricate interplay between mice and their gut fungal communities. The data offer valuable insight into the ecological adaptation in wild mice, emphasizing the complex and dynamic relationship between the murine hosts and their mycobiota.

**KEYWORDS**    wild mice, gut fungi, ecological adaptability, amplicon sequencing, whole-genome sequencing

T he gut microbiota is a highly complex ecosystem composed of bacteria, archaea, fungi, and other microorganisms (1). These microbiota are necessary for the development of the gut (2, 3), normal physiological and immunological functions (4, 5), and digestion of food (6). Understanding the factors contributing to variations in gut microbiota has gained more attention in the last few years. A comprehensive study comparing the gut microbiota of 60 mammalian species revealed marked gut microbiota differences within and among host species, with less differences detected within the same species compared to the differences among different species (7). Environmental factors, including array dietary habits (8) and physical separation (9) in genetically

Address correspondence to Xiao-Xuan Zhang, zhangxiaoxuan1988@126.com.

Ji-Xin Zhao, Hany M. Elsheikha, and Kai-Meng Shang contributed equally to this article. The author order was determined by contribution to research.

The authors declare no conflict of interest.

See the funding table on p. 14.

identical mice, can alter the composition and metabolism of the intestinal flora. In laboratory mice, distinct composition of gut microbiota is reported among 10 genetically distinct, inbred mouse lines, and the differences in the microbiota could be ameliorated via cohousing mice (10) and likely related to the cage effect (11, 12). Hence, unraveling the intricate composition and diversity of the gut microbiota can provide new insight into the microbial basis of ecological adaptation of species.

Previous studies have mainly focused on the role of gut bacteriota in the niche adaptation of animals (13–15). Bacteria play important roles in immune regulation, energy metabolism, and environmental adaptation (13, 16). However, we know very little about the extent of the differences between the fungal populations between different species of mice and whether variations in the fungal microbiota may relate to the adaptability of mice to their environmental conditions. Fungi, as eukaryotic organisms, colonize the gut of most mammals (17–19), collectively known as mycobiota. Although the gut fungal population constitutes <0.1% of microbiota in the gut (20), they produce an array of enzymes, peptides, depsipeptides, siderophores, and antimicrobial peptides which play roles in immune regulation and metabolic processes in the host (21–24). The gut fungi also play a role in maintaining microbial balance and preventing the overgrowth of harmful microorganisms in the intestine (25), as well as influencing the host's metabolism and energy balance (21, 26).

Laboratory mice are an established model organism extensively used in life science research. On the same token, wild mice offer an effective model to study the physiological functions of wild mammals and may provide insight relevant to humans (27). In this study, we selected three species of wild mice in addition to laboratory mice and utilized ITS1 amplicon sequencing technology to conduct an in-depth analysis of the spectrum of mouse fungal microbiota and elucidate the differences between wild and laboratory mouse species. By integrating whole-genome sequencing with culturomics, we conducted whole-genome sequencing of 48 fungi isolated from cultured strains and conducted in-depth investigations into their biological functions to explore the potential role of intestinal fungi in the genetic diversity of wild mice. This research enhances the functional gene repository of gut fungi and provides insights into the complex interactions between mice and their gut fungal communities, highlighting the genetic diversity of wild mice and their dynamic relationship with mycobiota.

## RESULTS

### Wild mice have richer gut mycobiota diversity

To understand differences in the mycobiota between wild and laboratory-reared mice, cecal contents from 81 mice were collected for ITS1 sequencing. On average, 66,988 reads of ITS were generated for each sample. A total of 2,912 ASVs were identified in the cecal fungal communities of the four species of mice (Table S3). The rarefaction curve analysis demonstrated that the curves for all samples reached a saturation plateau, indicating that the sequencing depth was sufficient to capture the richness and diversity of the mycobiota (Fig. 1A). Next, we compared the alpha diversity among cecal mycobiota. The observed ASV Richness and Shannon diversity were significantly higher in the three wild mouse species compared to C57BL/6J mice ($P < 0.001$). Specifically, Shannon diversity of *L. brandtii* was significantly higher than that of the other two wild mouse species ($P < 0.05$), and the Richness (Observed) was notably higher than that of the *M. fortis* (Fig. 1B and C). Bray-Curtis beta diversity revealed distinctive patterns among the four mouse species. PERMANOVA substantiated the significant differences in the composition of mycobiota among the four mouse species (PERMANOVA, coefficient of determination $R^2 = 0.3619$, $P < 0.001$; *M. fortis* vs C57BL/6J: $R^2 = 0.3820$, $P < 0.001$; *A. agrarius* vs C57BL/6J: $R^2 = 0.3615$, $P < 0.001$; *L. brandtii* vs C57BL/6J: $R^2 = 0.2252$, $P < 0.001$; *L. brandtii* vs *M. fortis*: $R^2 = 0.2603$, $P < 0.001$; *L. brandtii* vs *A. agrarius*: $R^2 = 0.2315$, $P < 0.001$; *M. fortis* vs *A. agrarius*: $R^2 = 0.2281$, $P < 0.001$; Fig. 1D).

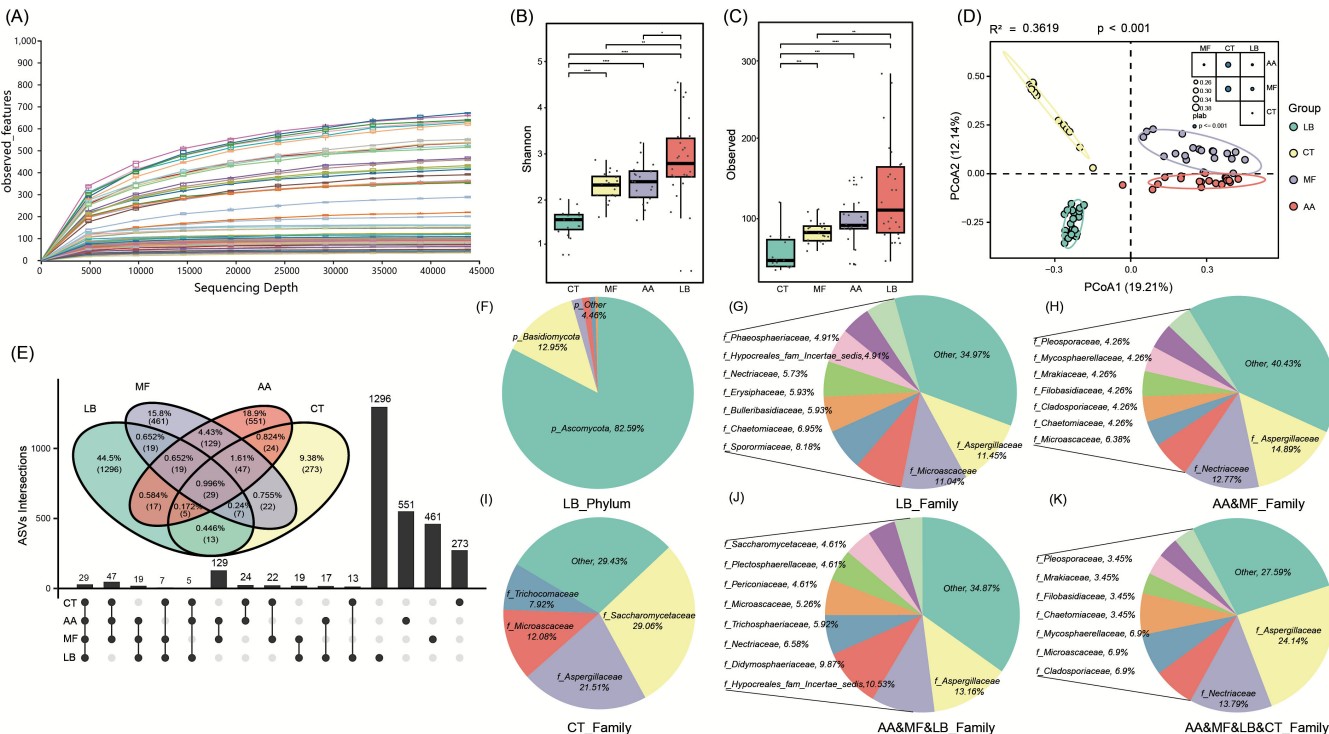

**FIG 1** Analysis of intestinal fungal community diversity in different mouse strains. (A) The rarefaction curve analysis for all samples. (B, C) Boxplots show the Shannon and Chao1 indices across all mouse groups, respectively. Wilcoxon rank-sum test: *P < 0.05; **P < 0.01, ***P < 0.001, ****P < 0.0001. (D) The scatter plot exhibits β-diversity of mycobiota across all mouse groups. Samples are shown at the first and second principal coordinates (PCoA1 and PCoA2), and the ratio of variance contributed by these two PCoAs is indicated. The central point within each group represents the mean coordinate value of the samples belonging to that specific group. Ellipsoids represent a 90% confidence interval surrounding each group. PERMANOVA results reveal the overall effect size of the four mouse species, and the effect sizes of each group pair are shown in the left-bottom panel. *P* values were calculated using the adonis test with 1,000 permutations in R. (E) Unique and common ASV distribution of the gut fungi in the four mouse species. (F, G) Distribution of ASVs specific to *Lasiopodomys brandtii* (LB) at the phylum and family level. (H) Distribution of ASVs shared by *Microtus fortis* (MF) and *Apodemus agrarius* (AA) at the family level. (I) Distribution of C57BL/6J (CT)-specific ASVs at the family level. (J) Distribution of ASVs common to three species of wild mice at the family level. (K) Distribution of ASVs shared by the four mouse species at the family level.

## Species-specific community structure of cecal mycobiota in mice

PCoA analysis results revealed significant differences in cecal fungal community structure among hosts. We explored how these differences manifest at different taxonomic levels, including species composition at the phylum and genus levels, as well as specific differences in species abundance between mouse groups. At the phylum level, *Ascomycota* dominates, constituting 71.72%–98.98% of the total reads (Fig. 2A; Table S4). Notably, *Basidiomycota* was significantly enriched in the wild mice, while *Mucoromycota* exhibited a significant enrichment in *L. brandtii* (Wilcoxon-rank sum, P < 0.05) (Fig. 2C through E; Table S5).

At the genus level, species groups are more sensitive to intergroup variation, thus more accurately reflecting differences in genus composition among different hosts. A total of 430 genera were identified in 81 samples, the top 5 by average relative abundance were *Aspergillus* (15.60%), *Paraphaeosphaeria* (9.76%), *Microascus* (9.63%), *Podospora* (6.64%), and *Nigrospora* (6.20%) (Fig. 2B; Table S4). The relative abundance of *Phaeosphaeride* and *Nigrospora* in the cecal mycobiota of *M. fortis* and *A. agrarius* was notably higher compared with the other two mouse species. Additionally, *Aspergillus* and *Microascus* exhibited a significantly higher relative abundance in C57BL/6J mice relative to the three wild mouse species (Fig. 2F through I; Table S5).

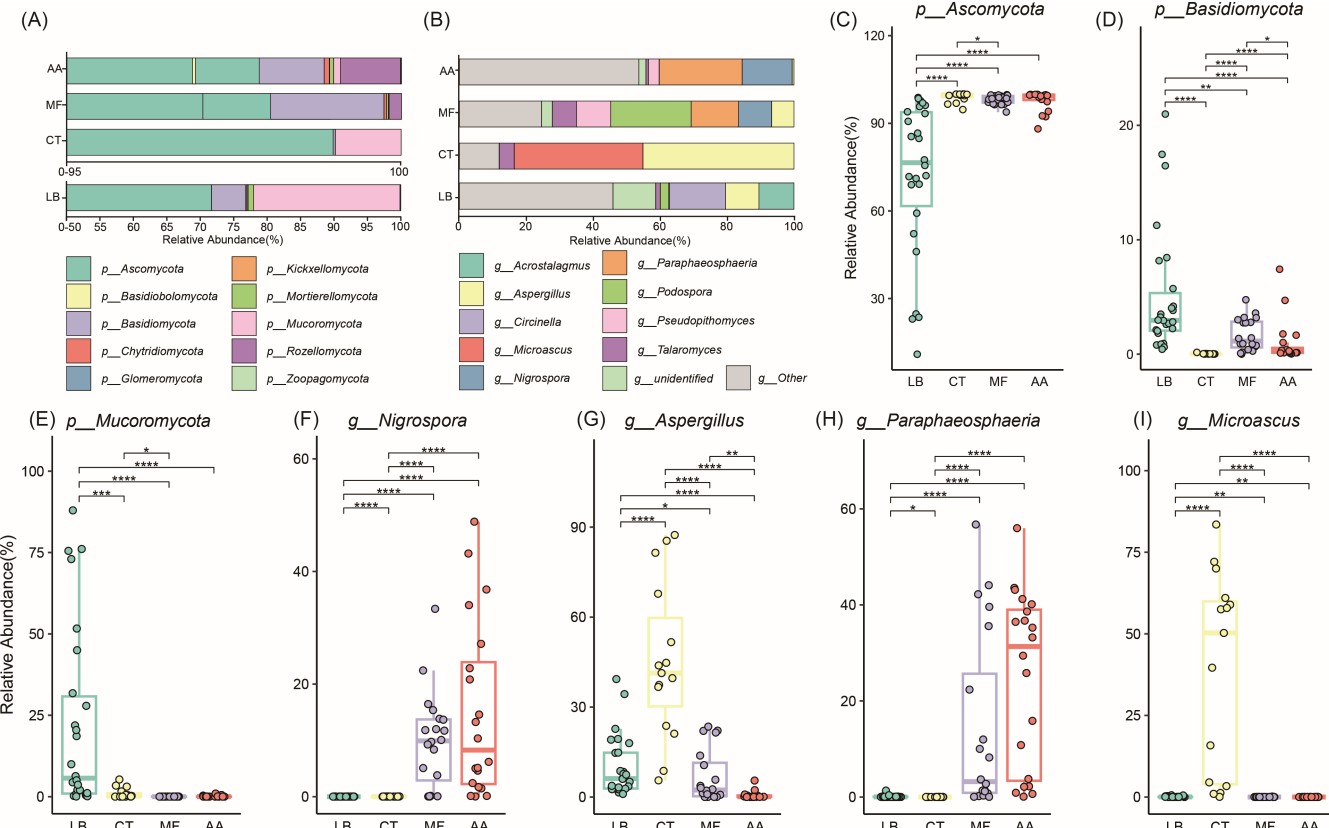

**FIG 2** Species composition analysis of intestinal fungal communities of different mouse strains. (A, B) Bar graphs depict the composition of the gut fungal population at the phylum and genus level, respectively. (C–E) Boxplots show the relative abundance of *Ascomycota*, *Basidiomycota,* and *Mucoromycotain* in different mouse groups. (F–I) Boxplots show the relative abundance of *Nigrospora*, *Aspergillus*, *Paraphaeosphaeria,* and *Microascus* in different mouse groups.

## Unique and shared ASVs between all mouse species

Exploring unique and shared fungal ASVs among different hosts enhances our understanding of fungal community diversity. In Fig. 1E, we analyzed the common ASVs, unique ASVs, and core ASVs. The results revealed a significantly higher number of unique fungal ASVs in the intestines of wild mice compared with C57BL/6J mice (*L. brandtii*, *n* = 1296; *A. agrarius*, *n* = 551; *M. fortis*, *n* = 461; CT, *n* = 273), reflecting the more diverse and rich fungal communities in wild mice. Among them, *L. brandtii* exhibited the highest number of unique ASVs, predominantly for *Ascomycota* and *Basidiomycota*, including genera such as *Blumeria*, *Aspergillus*, and *Metschnikowia* (Fig. 1F and G), followed by *A. agrarius* and *M. fortis*. Examining the shared ASVs, *A. agrarius* and *M. fortis* exhibited the most shared fungi, primarily belonging to *Ascomycota*, *Aspergillaceae*, and *Hypocreales_fam_Incertae_sedis* (Fig. 1H). The three groups of wild mice collectively shared 47 ASVs, dominated by *Aspergillaceae*, *Nectriaceae*, and *Microascaceae*. However, ASVs in C57BL/6J mice were mainly composed of *Saccharomycetaceae*, *Aspergillaceae,* and *Microascaceae* (Fig. 1I and J). Additionally, the four mouse species shared 29 ASVs, comprised mainly of the common *Aspergillaceae*, *Nectriaceae*, and *Microascaceae* (Fig. 1K), indicating that these fungi may constitute the core fungal community in the murine intestine (28).

## Cultivation and genome sequencing of the gut fungi

The high diversity of the wild murine cecal mycobiota combined with the scarcity of reference fungal genomes isolated from the intestine prompted us to carry out further characterization of these fungi. To achieve this objective, we isolated 428 fungal strains

from the cecal samples of the different mouse species. This was enabled through a combination of 10 different types of culturing media and isolation under both aerobic and anaerobic conditions. ITS1 gene-targeted Sanger sequencing was subsequently used to confirm 257 of the fungal isolates, which represented 3 phyla, 9 classes, 12 orders, 24 families, 36 genera, 53 species (Table S6).

We selected 48 strains for further whole-genomic analysis based on their phylogenetic diversity (i.e., ITS rDNA polymorphism) and morphological features of the strains (Fig. S1). We obtained a total of 48 assembled genomes for the cultivated gut fungi catalog (Table S7a). The genome sizes and G + C contents of genomes ranged from 11 to 86 Mbp (median 30 Mbp) and 30.41% to 58.80% (median 48.18%), respectively. Genome assembly quality was high (avg N50 10.038 to 2,740.886 kbp, median 825.876 kbp) as 44/48 genome assemblies had an N50 ≥ 100 kbp. The estimated genome completeness was 97.81% (ranging from 94.50% to 99.90%) (Table S7b).

We used single-copy core marker genes ($n$ = 172) to construct the phylogenetic landscape of the 48 fungal isolate genomes (Fig. 3). A commonly accepted criterion for defining species boundaries among prokaryotes is an ANI ranging between 95% and 96% (29). We used an ANI threshold of 95% to discriminate species based on a survey of the available fungal genomes in the NCBI database, which contains 11,510 high-quality genomes (Table S7c); similarly, the ANI threshold for genus boundary was set at 80%. The sequenced genomes were clustered into 2 phyla, 4 classes, 7 orders, 15 families, 18 genera, and 27 species (Table S7d). Four of these genomes represented unknown species with no reference genomes available.

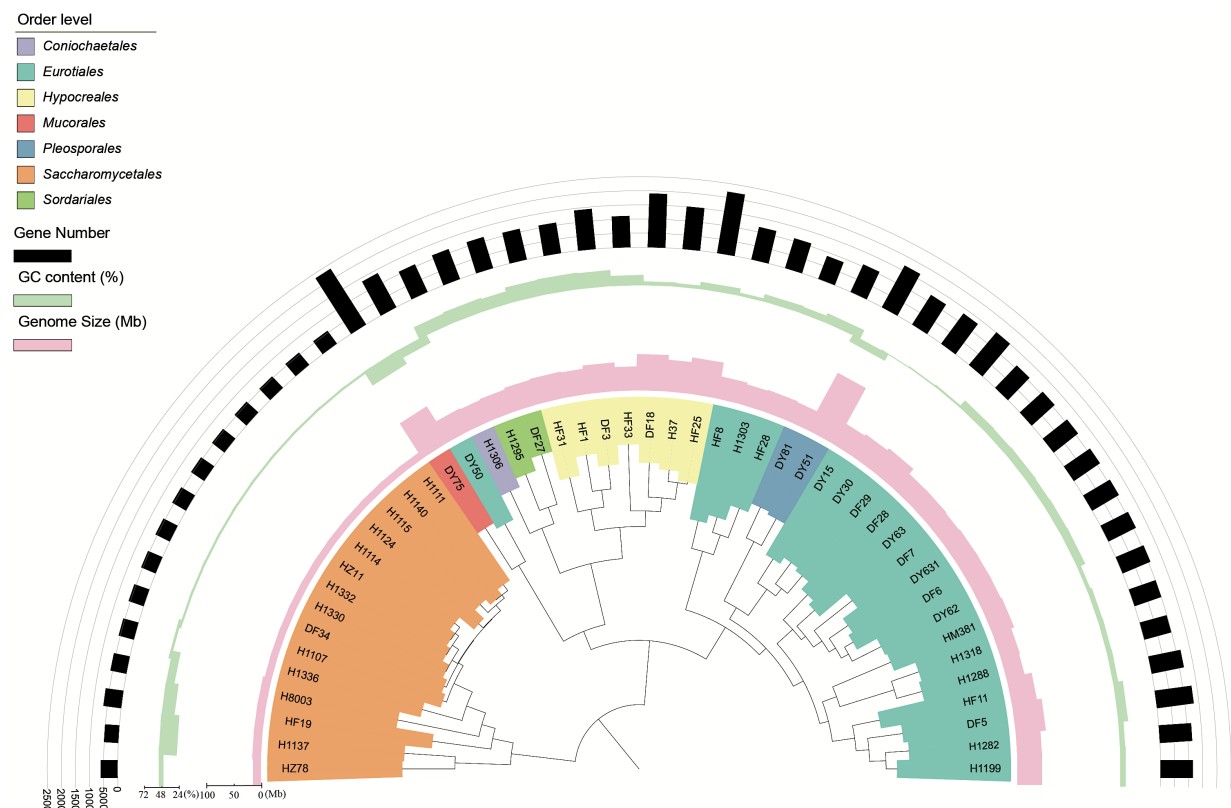

FIG 3   The phylogenetic relationship among 48 fungal genomes. The color code of each clade corresponds to the order-level classification of the genomes. The first, second, and third rings represent the genome size (Mb), the GC (%) content of the genome, and the number of genes annotated in the genome, respectively.

## Analysis of KEGG pathway integrity and COG profiling in fungal genomes

To better understand potential biological functions of the newly sequenced fungi, we assigned predicted genes against several reference databases. A total of 500,849 predicted genes (average 10,434 per genome, Fig. 4A through C; Table S8a) were detected. For each genome, the core functional pathways were reconstructed based on the completeness ratio of KEGG modules (Fig. 4D; see Table S8b for detail). We found that amino acid metabolism, carbohydrate metabolism, lipid metabolism, cofactor and vitamin metabolism, nucleotide metabolism, and energy metabolism have pathways in all genomes. The genomes of *Hypocreales* and *Pleosporales* exhibited more functional diversity compared with other fungal orders. Notably, essential components, such as the central carbohydrate metabolism citrate cycle (TCA cycle, Krebs cycle) (M00009), formaldehyde assimilation, xylulose monophosphate pathway (M00344) in energy metabolism, and the beta-oxidation process (M00087) and acylglycerol degradation (M00098) in fatty acid metabolism, are encoded in the majority of *Hypocreales* and *Pleosporales* genomes but are rarely found in other taxa (Fig. 4F through I).

To elaborate on the functions encoded by each strain, we categorized the KO annotations corresponding to each strain based on the KEGG pathway. The results showed that *Hypocreales* encodes the most antibiotic-related genes (including

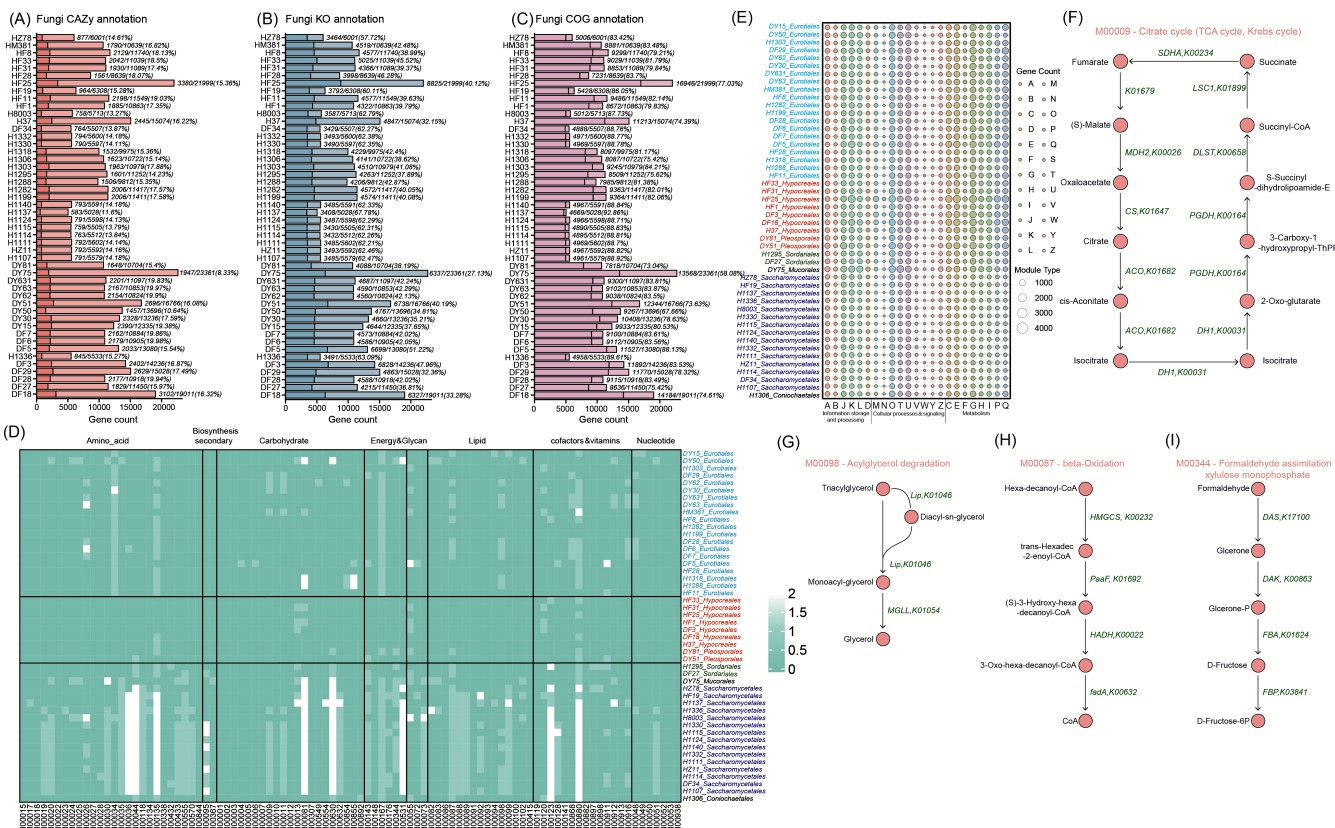

**FIG 4** Functional analysis of intestinal fungi. (A–C) Stacked histograms of the gene counts annotated CAZymes, COGs, and KOs for each genomes, respectively. Values shown are the numbers of the annotated and total genes. Dark colors represent annotation, and light colors denote no annotation. (D) The completeness ratio of KEGG metabolic modules for 48 fungal genomes. The color shows the median of the completeness ratio of the strains in each subclade: dark green, present; light green, largely present (only one enzyme not found). Only complete modules detected in at least 90% of the genome are shown. (E) A scatter plot shows gene counts, which annotated various functional units in each genome, including 23 COGs. The size of the points represents the number of genes. (F–I) The flow diagram shows the central carbohydrate metabolism citrate cycle (TCA cycle, Krebs cycle) (M00009), the beta-oxidation process (M00087) and acylglycerol degradation (M00098) in fatty acid metabolism, and formaldehyde assimilation, xylulose monophosphate pathway (M00344) in the pathways of energy metabolism. The nodes represent various compounds, the arrow lines indicate the direction of chemical reactions, and the green texts represent gene symbols. Visualization was performed in Adobe Illustrator CC 2020 with reference to KEGG map00020, map00071, map00561, and map00680, respectively.

monobactam biosynthesis, penicillin and cephalosporin biosynthesis, carbapenem biosynthesis, novobiocin biosynthesis, streptomycin biosynthesis, neomycin, kanamycin, and gentamicin biosynthesis) (Table S8c). We identified 400,890 genes that could be assigned to eggNOG categories (80.04% of the total annotated genes, on average 8351.88 per genome, Fig. 4E). Information storage and processing, cellular processes, and signaling, as well as metabolism, constituted 18.03%, 20.83%, and 33.16% of the total number of these genes, respectively. The functions of these fungi were mainly related to metabolism, including carbohydrate transport and metabolism (category G, 6.50%), amino acid transport and metabolism (category E, 5.94%), secondary metabolites biosynthesis, transport, and catabolism (category Q, 5.17%). Interestingly, *Hypocreales* that were significantly higher in the gut of wild mice than in the C57BL/6J mice encoded the most genes and unknown genes related to secondary metabolites biosynthesis, transport, and catabolism (Fig. S2B and Table S8d). Furthermore, we compared the conspecific genomes isolated from laboratory mice and wild mice. These genomes showed a high degree of consistency in structure and function (Fig. S2D and E).

## The functions of cecal fungi enhance their niche adaptability

Functions involved in the degradation of fermentable substrates (i.e., polysaccharides, proteins, and lipids) and biosynthesis of secondary metabolites play a key role in the ecological adaptation of fungi to the mammalian gut environment (30, 31). We, thus, examined the genes coding for CAZymes in the 48 genomes to assess their involvement in metabolizing polysaccharides. A total of 80,954 CAZymes were predicted, representing 16.04% of the total protein repertoire in the studied genomes (Table S8e). *Eurotiales*, *Hypocreales*, *Pleosporales*, and *Sordariales* strains encoded a higher proportion and more diverse repertoire of CAZymes, compared to other fungi (Fig. 5A). They expressed numerous plant cell wall degrading enzymes (PCWDEs) that act on cellulose, hemicellulose, starch, and pectin, implying their role in decomposing plant polysaccharides in the mouse gut, and supporting previous work on *Aspergillus* spp (32). In addition, we detected 10,321 protease genes and 12,323 lipase genes, representing 2.06% and 2.46% of the total protein repertoires, respectively (Fig. 5B and C). The enrichment analysis revealed that *Hypocreales* and *Pleosporales* species exhibit the highest lipid catabolic capacity and significant proteolytic capacity compared to other lineages (Table S8f and g). Finally, a total of 23,432 secondary metabolism gene clusters (SMGCs) were identified in the genomes, which were clustered into 1,539 families (32). *Eurotiales* and *Hypocreales* genomes exhibited the highest proportion of SMGCs compared with other orders; together they encoded 86.85% of the SMGC families. At the same time, we found that each species encodes a unique combination of SMGCs (Fig. 5D; Table S8h). This result agrees with previous data on the *Aspergillus* species (32), suggesting that horizontal gene transfer is a common mechanism in the evolution of fungal secondary metabolism.

## DISCUSSION

The gut microbial communities play key roles in mediating the host physiological functions (33, 34), growth and development including maturation of the immune system (35), and resistance to pathogens (36). Microbiota are particularly important in mediating the host environmental adaptation (9) because of their rapid responsiveness to environmental changes at the population and individual microbial lineage levels. However, the role of the variations in the gut fungal microbiota and their role in the adaptation of different mouse species to different ecological habitats remain largely unknown.

In this study, we used ITS1 amplicon sequencing to provide a comprehensive analysis of the composition and diversity of cecal fungal microbiota in three wild mouse species vs laboratory C57BL/6J mice. Our results revealed marked differences in the structure and diversity of cecal fungal communities among the different mouse species. The fungal communities exhibited higher diversity and abundance in *L. brandtii* compared to the other three mouse species. Additionally, *L. brandtii* had significantly higher abundance of *Ascomycota*, *Basidiomycota,* and *Mucoromycota* than the other three mouse species. This

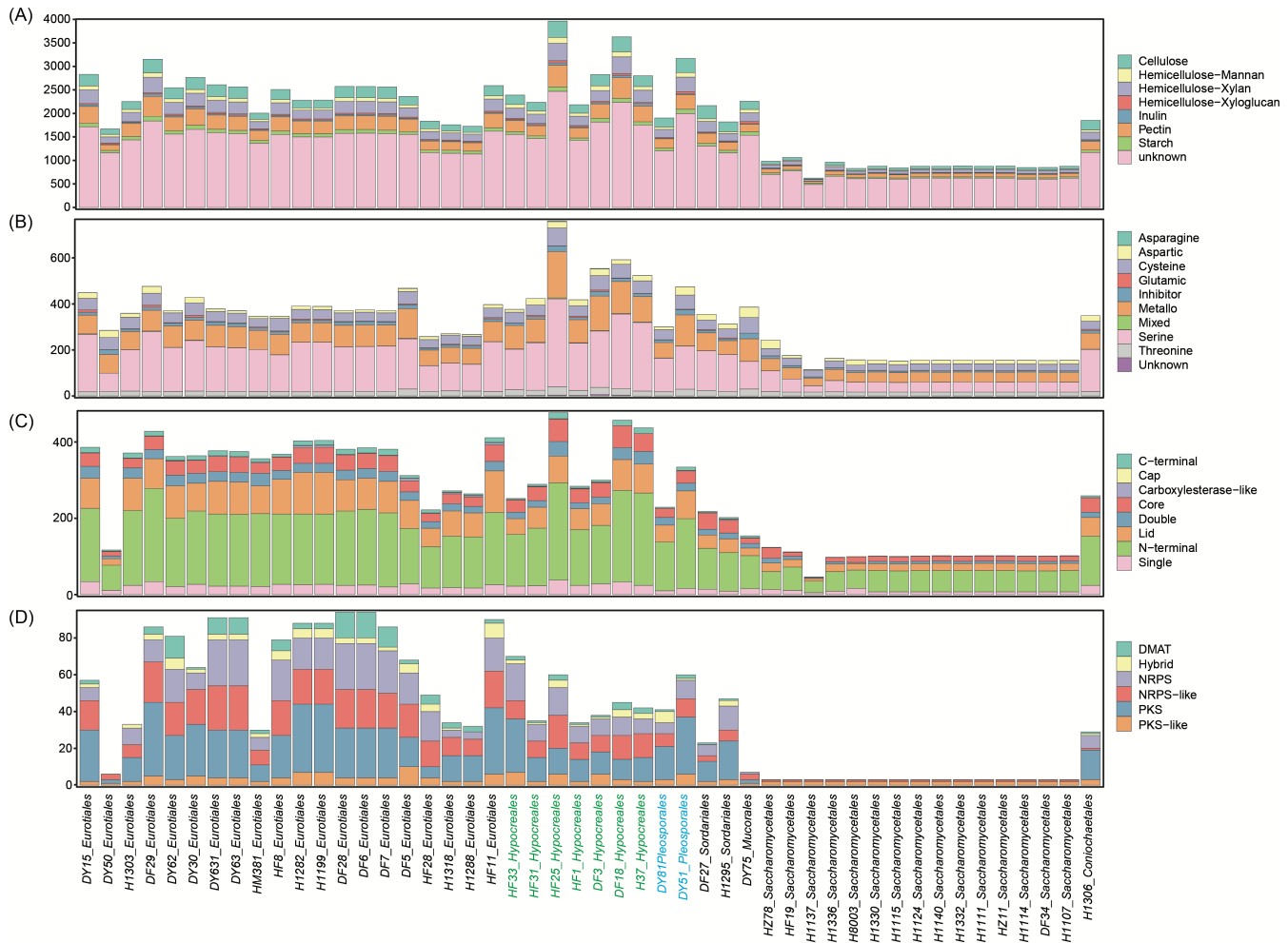

**FIG 5** Bar plots show the number of CAZymes (A), Proteases (B), Lipases (C), and SMGCs (D) of 48 fungal genomes. For CAZymes, the gene contents of seven target polysaccharides are shown. The lipases are categorized based on the lipase engineering database (http://www.led.uni-stuttgart.de/) (8). The SMGCs are classified by backbone enzyme types: DMAT, dimethylallyl transferase; NRPS, non-ribosomal peptide synthetase; PKS, polyketide synthase; Hybrid, the gene containing domains from NRPS and PKS backbones.

result may have a plausible explanation because *L. brandtii* were collected at a higher altitude than the other three mouse species, and the richness and diversity of gut microbial communities positively correlate with altitude (37, 38).

Overall, wild mice exhibited more abundance and diversity in the fungal communities compared to C57BL/6J mice. The increased richness and diversity of cecal fungal communities in wild mice may help them to adapt to their dynamic environmental and dietary conditions (39). The influence of the environment, including diet (24) and the interference of diet with host genetics (40), on shaping the abundance and composition of gut mycobiome in mice have garnered increased attention. A previous study showed that laboratory mice sourced from different animal vendors exhibit significant differences in the gut mycobiome which are markedly decreased after feeding mice the same diet for 8 weeks (24). In analogy to the previous findings, the present study showed that the cecal fungal communities had less difference between *A. agrarian* and *M. fortis* captured in the same area compared to the other two mouse species (*L. brandtii* and C57BL/6J). The limited diversity in the mycobioate of *A. agrarian* and *M. fortis* which share in the same environment may be attributed to their access to similar dietary source. Our results agree with others showing that gut mycobiome in healthy mice is highly variable and responds to disturbances such as changes in the environment and diet (24).

Considering the significant differences in the composition of cecal fungal microbiota in wild mice compared with C57BL/6J mice, we explored the functional role of the fungi in the adaptation of wild mice to their ecological environment. We identified fungal lineages with a positive impact on the fitness of wild mice. For example, *Paraphaeosphaeria* and *Nigrospora* were significantly more abundant in *M. fortis* and *A. agrarius*, captured during the dry season in the Dongting Lake area, compared to the other two mouse groups. *Paraphaeosphaeria* have anti-inflammatory and anti-cancer-related properties (41), and nigronapthaphenyl, extracted from the endophytic fungus *Nigrospora sphaerica*, has anti-bacteria, anti-cancer, anti-inflammatory, and α-glucosidase inhibitory activities (42). Therefore, *M. fortis* and *A. agrarius* survival under harsh environmental conditions (43) is potentially attributed to the health-promoting bioactive compounds secreted by their gut fungi. The relative abundance of *Aspergillus* in C57BL/6J mice was significantly higher than that of all wild mouse spp. Laboratory rodents typically live under controlled housing and husbandry conditions with a relatively standardized diet. *Aspergillus* is often used as a feed additive (44–46), which may increase the abundance of these fungi in the gut of laboratory mice (47). A previous study showed that jejunal fungal communities in mice can be altered by dietary intervention and hence susceptible to environmental influence (24). Hence, the results of the present and previous studies (24) suggest that differences in the gut mycobiome detected between mouse species feeding on different diets may underpin the variability in ecological adaptability in wild mice.

Variations in mouse gut mycobiome abundance and composition, in response envionemnetal and dietary factors including, can impact host metabolism and body energy homeostasis (24). Therefore, we performed KEGG pathway enrichment analysis of the genomes of 27 fungal species and detected, using comparative genomics, a high degree of structural and functional consistency among the same genomes isolated from different mouse species. Our results showed the versatile role of cecal fungi in modulating host metabolism. We found that all fungal species had near-complete central metabolism (i.e., glycolysis and tricarboxylic acid cycle), fatty acid metabolism (fatty acid biosynthesis and beta-oxidation), and nucleotide metabolism pathways. Core metabolic pathways like glycolysis and the tricarboxylic acid cycle aid in organic matter breakdown and energy generation (48), bolstering the host's adaptability to the environment. Fatty acids are important substances for energy storage and structural components of organisms (49). Fungal involvement in fatty acid synthesis and metabolism can influence the host lipid metabolism and composition, thus impacting environmental adaptation (50). Likewise, fungal participation in nucleotide pathways may affect host nucleic acid metabolism and gene expression, altering cell growth and functions (51). In addition, the metabolites produced by these metabolic pathways may affect the activity and function of the immune system (52, 53), regulating the host's ability to respond to environmental stress and pathogenic infections. The pathway modules within the intestinal microbiota exert a notable influence on the host's adaptation to the ecological habitats (15). COGs and KOs annotation analysis identified numerous genes distributed across functional categories, including carbohydrate transport and metabolism, amino acid transport and metabolism, and secondary metabolite biosynthesis, transport, and catabolism.

By combining the amplicon results (Tables S5 and Fig. S2A through C) with the functional analysis results, we found that the genomes of *Eurotiales*, *Hypocreales*, *Pleosporales*, and *Sordariales* were the most prevalent. These genomes included pathways enriched in tyrosine and tryptophan degradation, as well as nitrate assimilation. Moreover, these genomes had several catabolic enzyme genes, particularly those related to plant cell wall degrading enzymes (PCWDEs) and lipases. The abundance of secondary metabolite gene cluster (SMGC)-related genes indicates that they can produce a variety of secondary metabolites, including polyketide synthases (PKSs), non-ribosomal peptide synthetases (NRPSs), and terpene cyclases (TCs). PKSs and NRPSs play roles in synthesizing many antibiotics (54, 55), with NRPSs being important in the synthesis of agents with potential anti-cancer activities (56). These enzymatic pathways

may contribute to host resistance against pathogenic infections. TCs can be used to synthesize terpenoids (57), which have antimicrobial (58) and antioxidant (59) properties and play a role in maintaining microbial balance in the host's gut.

These results demonstrate the significance of the higher abundance of *Hypocreales* and *Pleosporales* in the three wild mice compared to C57BL/6J mice (Fig. S2B and C). The genomes of *Hypocreales* and *Pleosporales* exhibited marked functional diversity and were highly enriched in key metabolic pathways, integral to the host's adaptation to ecological environments (60), such as the central carbohydrate metabolism citrate cycle (TCA cycle, Krebs cycle) (M00009), formaldehyde assimilation, xylulose monophosphate pathway (M00344) in energy metabolism, among others. *Hypocreales* also encode many genes related to antibiotic synthesis and secondary metabolite biosynthesis, transport, and catabolism. These results support previous studies (24, 40), showing the effect of different mouse species living in different habitats on the abundance, composition and functions of the gut fungal microbiota and substantiate the role of mycobiota in influencing the host's metabolism and adaptive responses to its ecological milieu.

To sum up, this work revealed compositional variations in the cecal mycobiota among the different mouse species, with marked increases in the abundance and species diversity of the cecal fungi in wild mice compared with laboratory mice. We identified fungal lineages, such as *Phaeosphaeride* and *Nigrospora*, which have health beneficial effects and positive impact on the fitness of wild mice. Our data also revealed functional variations in the mycobiota in terms of phenotypic traits such as energy uptake from diet and defense against pathogens and a substantial presence of carbohydrate enzymes, metabolism-related COGs and KOs, and a notable potential for biosynthesis of antibiotics and secondary metabolites within the cecal mycobiota. The results suggest that variations in the fungal community composition may relate to the adaptability of mice to their environmental conditions. Our data form a foundation for research into the relationship between mycobiota and the murine host and their role in the adaptive mechanisms employed by different mouse species in different ecological habitats.

## MATERIALS AND METHODS

### Experimental design and sample collection

C57BL/6J mice (*n* = 15) and individuals from three wild mouse species (total *n* = 66) raised over 8 weeks in a controlled environment were selected. All C57BL/6J [purchased from (Beijing) Biotechnology Co., Ltd.] mice were housed in a specific pathogen-free facility with controlled environmental temperature and humidity (22 ± 2°C, 50% ± 10%), a 12:12 hour light/dark cycle, and an independent ventilation system, with food and water available *ad libitum*. The wild mice captured using peanuts or sunflower seeds as bait comprised 26 Brandt's voles (*Lasiopodomys brandtii*), 20 striped field mice (*Apodemus agrarius*), and 20 reed voles (*Microtus fortis*). Detailed information on the mouse species, sex, age, weight, sampling time, latitude and longitude of the sampling area, and altitude is shown in Table S1. All mice were transported to the laboratory, euthanized using carbon dioxide inhalation, followed by cervical dislocation. The ceca were immediately harvested post-euthanasia, and the cecal content was carefully collected for analysis. The cecal contents of each mouse were divided equally into two aliquots: one aliquot was stored at −80°C for DNA extraction and the other aliquot was used for culturomics analysis.

### DNA extraction and amplicon sequencing

The DNA was extracted from the cecal content using the TIANGEN stool DNA kit (TIANGEN Biotech Co., Ltd., China) according to the manufacturer's instructions. DNA was extracted three times independently from each cecal sample (81 samples in total) to reduce extraction bias. The quality of DNA was detected by 1% agarose gel electrophoresis. Fungal DNA was amplified using the universal primers pair ITS1F (5′-CTTGGTCATTT

AGAGGAAGTAA-3′) and ITS2R (5′-GCTGCGTTCTTCATCGATGC-3′) (61), targeting the ITS1 region. The PCR amplification was performed using Taq DNA polymerase (Thermo Fisher Scientific, Cat. No. EP0402, 5 U/µL). The PCR amplification protocol was as follows: an initial denaturation step at 95℃ for 3 min, followed by 35 cycles of denaturation at 95℃ for 30 s, annealing at 55℃ for 30 s, and elongation at 72℃ for 45 s, concluding with a final extension at 72℃ for 10 min. (Each time the extracted DNA was subjected to PCR amplification three times to ensure the reliability of the amplification.) The PCR amplification product was detected using 2% agarose gel electrophoresis, and the target fragment was subjected to extracted from the gel. The purified PCR products of each sample were independently barcoded and then pooled together for sequencing library construction. The TruSeq DNA Sample Prep Kit was used to construct the sequencing libraries, ensuring that each of the 81 samples was individually labeled with a unique barcode. The pooled, barcoded samples were subsequently prepared and processed according to the manufacturer's protocol for high-throughput sequencing. The library's concentration was quantified using Qubit, and high-throughput sequencing was subsequently performed on the NovaSeq 6000 platform.

## Bioinformatics and statistical analyses

Raw data were demultiplexed by the MiSeq Controller Software (Illumina Inc.). QIIME2 (v2023.5) was used for the downstream analysis (62). The demultiplexed ITS sequences were quality-filtered and grouped into amplicon sequence variants (ASVs) using DADA2 (63). Taxonomic classification of the representative ASVs was performed using feature-classifier plugin in QIIME2 based on the UNITE database (v8.2, 99%). Features present in only one sample were excluded. The number of ASVs was normalized via subsampling 44,052 sequences from each sample. The rarefaction curve analysis was performed using the diversity plugin. The Shannon (64) and Richness (65) indices were calculated using the abundance profiles of the features using "diversity" function in "vegan" R package (66). To assess the β-diversity, principal coordinate analysis (PCoA) was performed based on the Bray-Curtis distance, and the significance of differences between groups was determined using permutational multivariate analysis of variance (PERMANOVA) (66, 67). The Wilcoxon rank-sum test was performed to evaluate the level of significance in the difference in the diversity indices and abundance of taxa among the different groups (68). UpSet and venn diagrams were used to visualize the unique and shared ASVs across all groups using R package UpSetR (v1.4.0) (69) and VennDiagram (v1.7.3) (70). All other visualizations were generated using the ggplot2 package (v3.4.2) (71) in the R environment (v4.3.1) (72).

## Fungal isolation and identification

To account for the diverse nutritional needs of different fungal strains, we used a suite of 10 culture media, each with a different nutritional composition. Detailed information of the culture media was listed in the Table S2. All media were supplemented with penicillin 100 U/mL and streptomycin 100 µg/mL. In a sterile environment, each sample ($n = 81$) was diluted with phosphate-buffered saline (PBS) to achieve dilution factors of 40×, 400×, and 4,000×. Then, 1 mL of each suspension was inoculated onto agar plate containing a fungal culture medium, and the plates were incubated at 32℃ under both aerobic and anaerobic conditions until colonies appeared (73). [Each sample dilution (40×, 400×, 4,000×) is plated on 10 culture medium plates (3 dilution gradients × 10 culture media = 30 plates/sample).] To ensure that the isolated bacteria originate solely from the mouse intestine, the cecal contents were promptly diluted and inoculated into the culture medium. All experiments were conducted in a clean laboratory equipped with an independent filtration and ventilation system, and all sample collection consumables were disposable and sterile. The collection of fresh samples, their subsequent frozen storage, and incubation in a culture medium were all completed within two hours. After incubation for 2–14 days, various fungal colonies were observed on the cultured agar plates. In contrast, no fungal colonies were observed on the agar

plates used for culturing the sterile diluent buffer, which served as the negative control. Phenotypically distinct colonies from each incubated agar plate were transferred to the fresh corresponding medium for further purification. Subsequently, a single colony was selected and re-streaked onto the same type of medium to ensure further purification, following the same procedure. The purified fungal strains, intended for identification and storage, were inoculated into 5 mL Martin broth modified (MTB) medium at 32°C with shaking at 130 rpm. When the fungi grow to a suitable concentration (e.g., when a cloudy suspension of fungi can be observed in the liquid culture medium), take an aliquot of the culture medium and transfer it to a centrifuge tube. Centrifuge at 3,000–5,000 × *g* for 10–15 minutes. After centrifugation, carefully pour off the supernatant to obtain the fungal precipitate.

Clean fungal precipitates were collected for DNA extraction. Each isolated fungal strain was stored at −80°C in a 25% glycerol solution (74). The fungal genomic DNA was extracted using Fungi Genomic DNA Extraction Kit (Solarbio, China) for taxonomic identification and whole-genome sequencing, and the DNA products were stored at −20°C. The ITS1 region was amplified using the universal primers pair ITS1F (5′-TCCGTAG GTGAACCTGCGG-3′) and ITS2R (5′-GCTGCGTTCTTCATCGATGC-3′). The PCR amplification protocol was as follows: pre-denaturation at 98°C for 2 min, followed by denaturation at 98°C for 10 s, annealing at 55°C for 5 s, and extension at 72°C for 1 min. This cycle was repeated for 34 cycles, followed by a final extension at 72°C for 2 min. The PCR products were detected by 2% agarose gel electrophoresis. The PCR products that met the quality standards were used for Sanger sequencing (General Biological System Co., Ltd., China). The ITS1 region sequences were aligned against GenBank database using BLAST (https://blast.ncbi.nlm.nih.gov/Blast.cgi) to determine the taxonomic assignments. These strains were cataloged and stored in our laboratory repository. The ITS1 sequences and annotation files of all isolates have been uploaded to the Zenodo repository with access number 14784111 (https://doi.org/10.5281/zenodo.14784111).

## Whole-genome sequencing, genome assembly, and assessment of genome quality

A total of 48 cultured fungal strains were selected for whole-genome sequencing to obtain at least one representative genome for each fungal species of the isolated strains (after de-replication at 97% nucleotide similarity of their ITS rDNA sequences). A manual selection was performed based on the fungal physical appearances, such as color, texture, colony topography, and diffusible pigments. Sequencing libraries were prepared by using 1 µg of DNA as an input material and the NEBNext Ultra DNA Library Prep Kit for Illumina (New England BioLabs, USA), and index codes were added to attribute sequences to each sample. The Illumina NovaSeq platform was used for 2 × 150 bp paired-end sequencing. High-quality reads were filtered from the raw Illumina data by trimming the low-quality ($Q < 30$) bases at the end of reads and filtering "N"-containing, adapter contaminated, or short length (<90 bp) reads using fastp (75) with the parameter "-q 20 u 30 n 5 -y -Y 30 L 90—trim_poly_g". The *de novo* assembly of shotgun sequencing reads for each fungal isolate was performed using SPAdes (76) with k-mer parameters "21, 33, 55, 77" and mismatch correlation mode (--careful). The raw assembled sequences were processed by contig extension and scaffolding using SSPACE (77), and gaps were closed by GapCloser (78) through iterative runs until the best assembly result was achieved. The qualities of the fungal genome assemblies were assessed by quantifying their completeness using BUSCO (79). BUSCO used a set of 758 universal fungal single-copy orthologs (version 2020-09-10) to infer the completeness of a query fungus.

## Fungal genome taxonomic assignments and phylogenetic analysis

Fungal genome average nucleotide identity (ANI) was calculated using FastANI (29) with the default parameters. To explore species/genus demarcation in fungi, we systematically

generated pairwise ANIs for all fungal genomes obtained from NCBI. The accuracy of species/genus classification was critically evaluated with ANI thresholds of 80% and 95% for genus and species, respectively. Genomes of a species that did not match any reference genome in the NCBI using the species-level thresholds described above were considered novel species. For phylogenetic analysis, the prediction of fungal protein-coding genes was implemented using GeneMark-ES (v4.68_lic86) (80) with the parameters "--fungus --ES --min_contig 20000" first, leading to 500,849 predictive proteins. Orthologous gene clustering of the fungal genomes was performed to generate protein clusters using the MMseqs2 (v12.113e3) (81) algorithm with options "--min-seq-id 0.5c 0.9" (similarity 50% and minimum coverage threshold of 90% of the length of the shortest sequence) at the protein level. Among these protein families, we identified 172 protein families as single-copy protein markers. These markers must be present in all fungal genomes. The protein markers of each genome are combined into concatenated sequences to construct a phylogenetic tree. The specific process of constructing a phylogenetic tree is as follows: (i) Multiple Sequence Alignment and Trimming: Alignment Tool: Sequences were aligned using MAFFT (v7.475) with the --auto parameter to automatically select the optimal alignment strategy (82). No additional trimming was performed, as IQ-TREE's model selection and tree inference account for variable sites during analysis. (ii) Model Selection and Phylogenetic Tree Construction: Software: Phylogenetic analysis was conducted using IQ-TREE (v2.1.2) (83). Model Selection: The best-fit substitution model was automatically selected by IQ-TREE's built-in ModelFinder algorithm, which evaluates models using the Bayesian Information Criterion (BIC). Bootstrap Analysis: Branch support was assessed using ultrafast bootstrap approximation (UFBoot) with 1,000 replicates, a default setting in IQ-TREE that balances computational efficiency and robustness. (iii) Tree Visualization: Visualization Tool: The final maximum-likelihood tree was visualized and annotated using iTOL (Interactive Tree of Life, v5) (84).

## Gene functional annotation and analysis

The eggNOG-mapper (85) program was used to predict the functional repertoire of protein within 48 fungal genomes based on the eggnog (86) (evolutionary genealogy of genes: Nonsupervised Orthologous Groups, v5.0) databases. Kyoto Encyclopedia of Genes and Genomes (KEGG) annotation was performed by searching against the KEGG database (downloaded in April 2023) using DIAMOND (v2.1.8.162) (87) with a bit-score threshold of 60 and over 50% coverage. Each protein received an assignment to an eggnog and KEGG ortholog based on the best-hit gene in the corresponding database. The annotation of carbohydrate-active enzymes (CAZymes), proteases, and lipases for 48 fungal genomes was performed using the CAZy, downloaded in September 2023 (88), MEROPS (89), and Lipase Engineering Database (LED, v4.0) (90) databases, respectively. PCWDEs were determined using the CAZyme-based ranking of fungi (CBRF) (91). Annotation of the secondary metabolism gene clusters and identification of SMGC families in fungal genomes were carried out as previously described (32). A circular comparative genome map was constructed for conspecific genomes isolated from different hosts using the BLAST Ring Image Generator (BRIG, v 0.95) (92) with default parameters.

## ACKNOWLEDGMENTS

We thank Dr. Falk Hildebrand from Quadram Institute for the useful comments and constructive suggestions on an early draft of the manuscript.

The study was supported by the National Natural Science Foundation of China (Grant No. 32170538).

J.X.Z. performed experiments and data analysis and wrote the first draft of the manuscript. H.M.E. guided experiments and participated in editing the manuscript. K.M.S. performed experiments and participated in editing the manuscript. J.W.S. and

Y.J.W. performed experiments. Y.Q. and Z.Y.Z. collected the samples. H.M. performed experiments. X.X.Z. guided experiments and data analysis and helped develop the project and provided key reagents and funding for the project and participated in editing the manuscript.

## AUTHOR AFFILIATIONS

[1]College of Veterinary Medicine, Qingdao Agricultural University, Qingdao, Shandong, China
[2]Faculty of Medicine and Health Sciences, School of Veterinary Medicine and Science, University of Nottingham, Loughborough, United Kingdom
[3]College of Veterinary Medicine, Jilin Agricultural University, Changchun, Jilin, China

## AUTHOR ORCIDs

Ji-Xin Zhao  http://orcid.org/0009-0006-4789-7259
Hany M. Elsheikha  http://orcid.org/0000-0003-3303-930X
Xiao-Xuan Zhang  http://orcid.org/0000-0002-9470-2892

## FUNDING

| Funder | Grant(s) | Author(s) |
|---|---|---|
| National Natural Science Foundation of China | Grant No. 32170538 | Xiao-Xuan Zhang |

## DATA AVAILABILITY

The amplicon sequencing and whole-genomic shotgun sequencing data of this study have been deposited in the National Center for Biotechnology Information (NCBI) PRJNA1066080. This includes 81 raw amplicon sequencing data sets from mouse cecal microbiota, available via the SRA accessions SRR27603979–SRR27604059. Forty-eight high-quality fungal genome assemblies derived from whole-genome sequencing, accessible through BioSample accessions SAMN39484350–SAMN39484397. Additionally, the 48 high-quality fungal genome assemblies generated from whole-genome sequencing have been deposited and are publicly accessible through the Zonedo database. Accession number: 14939226 (https://doi.org/10.5281/zenodo.14939226). All other data supporting the findings of this study are available in the paper and supplemental materials or from the corresponding author(s) upon request.

## ETHICAL APPROVAL

All animal procedures were approved by the Qingdao Agriculture University Research Ethics Committee. All animal experimental procedures were maintained according to the Biosecurity Law of the People's Republic of China.

## ADDITIONAL FILES

The following material is available online.

### Supplemental Material

**Figure S1 (Spectrum02840-24-s0001.pdf).** Strain morphology of the cultured fungi.
**Figure S2 (Spectrum02840-24-s0002.pdf).** Species composition at order level and comparative genome analysis.
**Table S1 (Spectrum02840-24-s0003.xlsx).** Statistics of sample information.
**Table S2 (Spectrum02840-24-s0004.xlsx).** Information about the culture media.
**Table S3 (Spectrum02840-24-s0005.xlsx).** Abundance of 2,912 fungal ASVs.
**Table S4 (Spectrum02840-24-s0006.xlsx).** Fungus dominant phylum and genus percentage.

**Table S5 (Spectrum02840-24-s0007.xlsx).** Fungus dominant phylum, order, and genus Wilcox test.

**Table S6 (Spectrum02840-24-s0008.xlsx).** Information of 257 fungal isolates, including ITS1 gene sequences and taxonomy.

**Table S7 (Spectrum02840-24-s0009.xlsx).** Information about the fungal genome assembly, quality of the fungal genome assembly, the reference genome, and ANI.

**Table S8 (Spectrum02840-24-s0010.xlsx).** Details about functional annotation.

## Open Peer Review

**PEER REVIEW HISTORY (review-history.pdf).** An accounting of the reviewer comments and feedback.

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
