## [Reviewer comments · Microbiology Spectrum]

Microbiology Spectrum

Investigation of the genetic diversity of gut mycobiota of the wild and laboratory mice

Ji-Xin Zhao, Hany M. Elsheikha, Kai-Meng Shang, Jin-Wen Su, Yong-Jie Wei, Ya Qin, Zi-Yu Zhao, He Ma, and Xiao-Xuan Zhang

Corresponding Author(s): Xiao-Xuan Zhang, Qingdao Agricultural University

Review Timeline:

Submission Date:	November 5, 2024
Editorial Decision:	January 18, 2025
Revision Received:	February 10, 2025
Editorial Decision:	February 27, 2025
Revision Received:	March 4, 2025
Accepted:	March 5, 2025

Editor: Renee Arias

Reviewer(s): The reviewers have opted to remain anonymous.

Transaction Report:

DOI: <https://doi.org/10.1128/spectrum.02840-24>

Re: Spectrum02840-24 (Investigation of the role of gut mycobiota of the wild and laboratory mice in environmental adaptation)

Dear Prof. Xiao-Xuan Zhang:

Thank you for the privilege of reviewing your work. Below you will find my comments, instructions from the Spectrum editorial office, and the reviewer comments.

Revision Guidelines

Sincerely,
Renee Arias
Editor
Microbiology Spectrum

Reviewer #1 (Public repository details (Required)):

The PRJNA1066080 bio project in NCBI has only 81 fungi ITS Raw sequence reads. However, they have to deposit the processed amplicon reads and raw reads used for genome assembly. They also have to deposit fasta files of 48 whole genome assemblies in NCBI and provide the accession numbers in the manuscript.

Reviewer #1 (Comments for the Author):

Overall, the authors have made a very good effort. However, I have the following queries.

Line 96: Please correct the spelling of "ecal".

Line 104: Figure 1 B-C only indicates what one and two stars indicate in terms of significance level. Please add what 3, 4, etc. stars indicate. Also, more significant outliers are present in Figure 1b and C LB mice. How do you explain this? Also, a good number of outliers are present in Figure 2 C-I. Overall, these outliers could affect the statistical analysis. How do you explain this?

Line 151: Please correct the spelling of "combination"

Line 350: Overall, please add additional metadata (age, length, weight, date of collection etc.) about mice if available. Also provide additional details of laboratory mice husbandry (food, housing conditions like temperature, humidity etc.).

Line 370 - 387: Please provide references for R packages used.

Line 456: Any trimming of the alignment was performed before phylogenetic analysis. How did you select the best for phylogenetic analysis and specify the models used? Also determine the number of bootstraps and how the tree was visualized (software?).

Reviewer #2 (Public repository details (Required)):

The genomes of the 48 isolates sequenced need to be publicly available, and I would suggest that also the 48 isolates be placed in a public collection.

Reviewer #2 (Comments for the Author):

The manuscript: "Investigation of the role of gut mycobiota of the wild and laboratory mice in environmental adaptation" describes the cecal mycobiota of four species of mice from different environments as well as the genome sequencing of 48 fungal isolates obtained from the cecal samples. The manuscript includes a thorough analysis of comparisons across samples. My main concerns are as follows:

Title: I suggest removing "...in environmental adaptation" from the title; also replacing "...role..." for "genetic diversity". To refer to adaptation, the experiments should have been designed differently, besides, as the authors mentioned, the gut mycobiota is only 0.1% of the gut microbiota, thus, how can mycobiota be ascribed a "role" while ignoring the 99% of other microorganisms that were not quantified. The experiments reflect differences in gut mycobiome that are due to: Host species, Host diet and Host environment, do not show direct evidence that mice with different gut fungal diversity can "adapt" differently if given an environmental pressure.

Data availability: Bioproject PRJNA1066080 listed in the manuscript refers to "81 fungi ITS Raw Data" with only 3 GB of information. The 48 sequenced genomes need to be uploaded to a public Database (NCBI, EMBL, DNA DBJ) and accession numbers provided in the manuscript.

Methods: This section lacks clarity and details, it does not say how many DNA extractions and PCR reactions were done from cecal samples, how many Libraries were prepared for sequencing, number of samples plated; in addition, references are lacking for several methods used.

Other comments:

L26: need to provide species of laboratory mouse used

L34: replace "and" for comma in "...carbohydrate metabolism and fatty acid metabolism..."

L60: remove "The" before environmental and before dietary

L74: change "array" to "an array"

L79: use singular for "organism"

L82: replace "additional" for "addition"

L84: please remove "of"

L39, L88, and L90: For the reasons explained at the beginning of this review, I suggest downplaying the term ecological adaptations, in my opinion the experiments reflect differences in gut mycobiome that are due to: Host species, Host diet and Host environment (including geographic areas), do not show direct evidence that mice with different gut fungal diversity can "adapt" differently if given an environmental pressure. A different type of experiment would need to be designed to answer questions on adaptation.

L96: replace "ecal" for "cecal"

L108, L110, L110, L135, L138, L139, and rest of the manuscript: replace "agrariu" for "agrarius", and correct in L257.

L119: remove "the" before *L. brandtii*

L126: remove "the" before *M. fortis*

L144-146: can the authors elaborate on the AVS1211 that was present 34723 in sample CT7, and how this large number could have affected the analysis?

L151: replace "specie" for "species"

L170: replace "." For ", " after Table 9a).

L350: Supplementary Table 1 shows altitude "0" meters for C57BL/6J, is this correct? Is a laboratory at the sea level?

L358: replace purity for "quality"

L360-361: need to provide the Reference where the primer sequences were obtained
L361: Need to provide here and name/brand of polymerase used
L366: replace "gelation recovery" for "extracted from the gel"
L366: refers to a single library being prepared and sequenced
L373: the word "denoised" does not exist, please use common terminology for cleaning/curating sequences
L378, L380, L382: Need reference for Shannon and Richness indices, for PCoA analysis, Wilcoxon test,
L391: replace "supplemented" for "supplemented with"
L392: the antibiotics are listed using different units (units, micrograms), please double check if this was intended
L396-397: needs more clarity detailing what was done
L404: please use capital letter for Martin
L406: please explain how the precipitates were obtained
L153, L416: replace "sanger" for "Sanger"
L419: isolates need to be placed in a public database to make them accessible to the scientific community, please provide Accession numbers
L447: It is possible that PCR could have generated hybrid amplicons? Please consider.

February 7, 2025

Dr. Renee Arias

Co-Editor-in-chief

Microbiology Spectrum

Re: Revised Manuscript ID Spectrum02840-24R1

Dear Dr. Renee Arias

On behalf of all authors, I would like to thank the Editor and two reviewers for their comments and constructive suggestions on our **manuscript ID Spectrum02840-24**. We have significantly revised the manuscript based on the reviewers' comments, corrected all errors, added experimental details, and supplemented the data. In the next section, we will detail our point-by-point responses to specific comments and suggestions.

Responses to the comments and suggestions of reviewer #1:

Reviewer #1 (Public repository details (Required)):

Comment: The PRJNA1066080 bio project in NCBI has only 81 fungi ITS Raw sequence reads. However, they have to deposit the processed amplicon reads and raw reads used for genome assembly. They also have to deposit fasta files of 48 whole genome assemblies in NCBI and provide the accession numbers in the manuscript.

Authors' response: Thank you very much for your and constructive suggestions. We have publicly released all data under BioProject PRJNA1066080. This includes: 81 raw amplicon sequencing datasets from mouse cecal microbiota, available via the SRA accessions SRR27603979 – SRR27604059. 48 high-quality fungal genome assemblies derived from whole genome sequencing, accessible through BioSample accessions SAMN39484350 – SAMN39484397." And we have added this information to the manuscript.

Reviewer #1 (Comments for the Author):

Overall, the authors have made a very good effort. However, I have the following queries.

Authors' response: We thank Reviewer #1 very much for his/her favourable comments and constructive suggestions on our manuscript.

We have addressed all the suggestions and queries provided by the esteemed reviewer as indicated in the revised manuscript and in our response to the individual comments below.

Comment: Line 96: Please correct the spelling of "ecal".

Authors' response: Thank you very much for your and constructive suggestions. We have changed ecal to cecal in the manuscript.

Comment: Line 104: Figure 1 B-C only indicates what one and two stars indicate in terms of significance level. Please add what 3, 4, etc. stars indicate. Also, more significant outliers are present in Figure 1b and C LB mice. How do you explain this? Also, a good number of outliers are present in Figure 2 C-I. Overall, these outliers could affect the statistical analysis. How do you explain this?

Authors' response: Thank you very much for your constructive suggestions. We have added the content indicated by 3 stars and 4 stars in the manuscript, i.e. ***, $p < 0.001$, ****, $p < 0.0001$.

Explanation for Outliers:

The outliers in LB mice likely reflect true biological variation in fungal alpha diversity due to individual differences in foraging behavior, microhabitat use, or transient fungal exposures in the wild. For example, some LB mice may have consumed fungi-rich food (e.g., decaying plants) prior to sampling, leading to higher diversity. Wild mice are not genetically or behaviorally uniform, and their gut mycobiota is shaped by dynamic environmental interactions. These outliers are ecologically meaningful and highlight the importance of individual heterogeneity in natural populations.

Host-Specific Fungal Colonization: Certain individuals may harbor transient or specialized fungi due to unique dietary intake (e.g., a wild mouse consuming a rare

mushroom species).

We used nonparametric tests (Wilcoxon rank sum) for comparisons between groups, which are less sensitive to outliers than parametric tests.

Despite outliers, the overall trends (e.g., higher alpha diversity in wild mice vs. lab mice) remained statistically significant ($p < 0.01$), supporting the robustness of our conclusions.

Comment: Line 151: Please correct the spelling of "combimation"

Authors' response: Thank you very much for your and constructive suggestions. We have changed combimation to combination in the manuscript.

Comment: Line 350: Overall, please add additional metadata (age, length, weight, date of collection etc.) about mice if available. Also provide additional details of laboratory mice husbandry (food, housing conditions like temperature, humidity etc.).

Authors' response: Thank you very much for your and constructive suggestions. We provide the age, weight, and sampling time of mice in Supplementary Table 1, but we did not record the length of wild mice due to the harsh field conditions. We provide detailed information of experimental mice in the manuscript: All C57BL/6J (purchased from (Beijing) Biotechnology Co., Ltd.) mice were housed in a specific pathogen-free facility with controlled environmental temperature and humidity ($22 \pm 2^\circ\text{C}$, $50\% \pm 10\%$), a 12:12 h light/dark cycle, and an independent ventilation system, with food and water available ad libitum.

Comment: Line 370 - 387: Please provide references for R packages used.

Authors' response: Thank you very much for your comments and suggestions. We have added the relevant references to the manuscript.

Comment: Line 456: Any trimming of the alignment was performed before phylogenetic analysis. How did you select the best for phylogenetic analysis and specify the models used? Also determine the number of bootstraps and how the tree was visualized(software?).

Authors' response: Thank you for raising these important points. Below are the detailed steps of our phylogenetic analysis workflow:

1. Multiple Sequence Alignment and Trimming:

Alignment Tool: Sequences were aligned using MAFFT v7.475 with the --auto parameter to automatically select the optimal alignment strategy (doi: 10.1093/molbev/mst010). No additional trimming was performed, as IQ-TREE' s model selection and tree inference account for variable sites during analysis.

2. Model Selection and Phylogenetic Tree Construction:

Software: Phylogenetic analysis was conducted using IQ-TREE v2.1.2 (doi: 10.1093/molbev/msu300). Model Selection: The best-fit substitution model was automatically selected by IQ-TREE' s built-in ModelFinder algorithm, which evaluates models using the Bayesian Information Criterion (BIC). Bootstrap Analysis: Branch support was assessed using ultrafast bootstrap approximation (UFBoot) with 1000 replicates, a default setting in IQ-TREE that balances computational efficiency and robustness.

3. Tree Visualization:

Visualization Tool: The final maximum-likelihood tree was visualized and annotated using iTOL (Interactive Tree of Life, v5) (DOI: 10.1093/nar/gkab301). And I have added the software used for visualization in the manuscript.

Responses to the comments and suggestions of reviewer #2:

Reviewer #2 (Public repository details (Required)):

Comment: The genomes of the 48 isolates sequenced need to be publicly available, and i would suggest that also the 48 isolates be placed in a public collection.

Authors' response: Thank you very much for your and constructive suggestions. We have publicly released all data under BioProject PRJNA1066080. This includes: 81 raw amplicon sequencing datasets (targeting fungal ITS regions) from mouse cecal microbiota, available via the SRA accessions SRR27603979 – SRR27604059. 48 high-quality fungal genome assemblies derived from whole genome sequencing, accessible through BioSample accessions SAMN39484350 – SAMN39484397." And we have added this information to the manuscript.

Reviewer #2 (Comments for the Author):

The manuscript: "Investigation of the role of gut mycobiota of the wild and laboratory mice in environmental adaptation" describes the cecal mycobiota of four species of mice from different environments as well as the genome sequencing of 48 fungal isolates obtained from the cecal samples. The manuscript includes a thorough analysis of comparisons across samples.

Authors' response: We thank Reviewer #2 very much for his/her favourable comments and constructive suggestions on our manuscript.

We have addressed all the suggestions and queries provided by the esteemed reviewer as indicated in the revised manuscript and in our response to the individual comments below.

My main concerns are as follows:

Comment: Title: I suggest removing "...in environmental adaptation" from the title; also replacing "...role..." for "genetic diversity". To refer to adaptation, the experiments should have been designed differently, besides, as the authors mentioned, the gut mycobiota is only 0.1% of the gut microbiota, thus, how can mycobiota be ascribed a "role" while ignoring the 99% of other microorganisms that were not quantified. The experiments reflect differences in gut mycobiome that are due to: Host species, Host diet and Host environment, do not show direct evidence that mice with different gut fungal diversity can "adapt" differently if given an environmental pressure.

Authors' response: Thank you very much for your suggestion. Based on your suggestion, we have changed the title to Investigation of the genetic diversity of gut mycobiota of the wild and laboratory mice

Comment: Data availability: Bioproject PRJNA1066080 listed in the manuscript refers to "81 fungi ITS Raw Data" with only 3 GB of information. The 48 sequenced genomes need to be uploaded to a public Database (NCBI, EMBL, DNA DBJ) and accession numbers provided in the manuscript.

Authors' response: Thank you very much for your and constructive suggestions. We have publicly released all data under BioProject PRJNA1066080. This includes: 81

raw amplicon sequencing datasets (targeting fungal ITS regions) from mouse cecal microbiota, available via the SRA accessions SRR27603979 – SRR27604059. 48 high-quality fungal genome assemblies derived from whole genome sequencing, accessible through BioSample accessions SAMN39484350 – SAMN39484397." And we have added this information to the manuscript.

Comment: Methods: This section lacks clarity and details, it does not say how many DNA extractions and PCR reactions were done from cecal samples, how many Libraries were prepared for sequencing, number of samples plated; in addition, references are lacking for several methods used.

Authors' response: Thank you very much for your suggestion. We have supplemented this information in the manuscript.

About DNA extraction, PCR reaction and library construction

Each cecal sample (81 samples in total) was subjected to 3 independent DNA extractions (to reduce extraction bias). Each extracted DNA was subjected to 3 repeated PCR amplifications (to ensure the reliability of amplification). The purified PCR products of each sample were independently barcoded and then pooled together for sequencing library construction. The TruSeq™ DNA Sample Prep Kit was used to construct the sequencing libraries, ensuring that each of the 81 samples was individually labeled with a unique barcode. The pooled, barcoded samples were subsequently prepared and processed according to the manufacturer's protocol for high-throughput sequencing.

About how many samples were cultured

A total of 81 samples were cultured for culturing, and each sample dilution ($40\times$, $400\times$, $4000\times$) was inoculated on 10 culture plates (3 dilution gradients \times 10 culture plates = 30 plates/sample).

We have added relevant references to the article. Thank you again for your valuable comments.

Other comments:

Comment: L26: need to provide species of laboratory mouse used

Authors' response: Thank you very much for your and constructive suggestions. We

provide detailed information of experimental mice in the manuscript: All C57BL/6J mice were purchased from (Beijing) Biotechnology Co., Ltd., and the average weight of mice was 21 g when samples were collected. All C57BL/6J mice were housed in a specific pathogen-free facility with controlled environmental temperature and humidity ($22 \pm 2^\circ \text{C}$, $50\% \pm 10\%$), a light/dark cycle of 12:12 h, and an independent ventilation system, and food and water were available ad libitum.

Comment: L34: replace "and" for comma in "...carbohydrate metabolism and fatty acid metabolism..."

Authors' response: Thank you very much for your and constructive suggestions. We have revised this in the manuscript.

Comment: L60: remove "The" before environmental and before dietary

Authors' response: Thank you very much for your and constructive suggestions. We have revised this in the manuscript.

Comment: L74: change "array" to "an array"

Authors' response: Thank you very much for your and constructive suggestions. We have revised this in the manuscript.

Comment: L79: use singular for "organism"

Authors' response: Thank you very much for your and constructive suggestions. We have revised this in the manuscript.

Comment: L82: replace "additional" for "addition"

Authors' response: Thank you very much for your and constructive suggestions. We have revised this in the manuscript.

Comment: L84: please remove "of"

Authors' response: Thank you very much for your and constructive suggestions. We have revised this in the manuscript.

Comment: L39, L88, and L90: For the reasons explained at the beginning of this review, I suggest downplaying the term ecological adaptations, in my opinion the experiments reflect differences in gut mycobiome that are due to: Host species, Host diet and Host environment (including geographic areas), do not show direct evidence that mice with different gut fungal diversity can "adapt" differently if given an

environmental pressure. A different type of experiment would need to be designed to answer questions on adaptation.

Authors' response: Thank you very much for your suggestion. We have downplayed the concept of environmental adaptation in the article and changed it to genetic diversity according to your previous suggestion.

Comment: L96: replace "ecal" for "cecal"

Authors' response: Thank you very much for your and constructive suggestions. We have revised this in the manuscript.

Comment: L108, L110, L110, L135, L138, L139, and rest of the manuscript: replace "agrariu" for "agrarius", and correct in L257.

Authors' response: Thank you very much for your and constructive suggestions. We have revised this in the manuscript.

Comment: L119: remove "the" before L. brandtii

Authors' response: Thank you very much for your and constructive suggestions. We have revised this in the manuscript.

Comment: L126: remove "the" before M. fortis

Authors' response: Thank you very much for your and constructive suggestions. We have revised this in the manuscript.

Comment: L144-146: can the authors elaborate on the AVS1211 that was present 34723 in sample CT7, and how this large number could have affected the analysis?

Authors' response: Thank you very much for your question. During the analysis of this study, we have tried to remove the outlier and reanalyzed the results. The analysis results did not affect the conclusion of this study, but in order to ensure the reproducibility of this study, we still uploaded the original data table.

Comment: L151: replace "specie" for "species"

Authors' response: Thank you very much for your and constructive suggestions. We have revised this in the manuscript.

Comment: L170: replace "." For "," after Table 9a).

Authors' response: Thank you very much for your and constructive suggestions. We have revised this in the manuscript.

Comment: L350: Supplementary Table 1 shows altitude "0" meters for C57BL/6J, is this correct? Is a laboratory at the sea level?

Authors' response: Thank you very much for your and constructive suggestions. Our laboratory is located in Qingdao. We re-measured the altitude of the laboratory and determined that it is 12.2 meters above sea level. We have also made some changes in Supplementary Table 1.

Comment: L358: replace purity for "quality"

Authors' response: Thank you very much for your and constructive suggestions. We have revised this in the manuscript.

Comment: L360-361: need to provide the Reference where the primer sequences were obtained

Authors' response: Thank you very much for your and constructive suggestions. We have provided a reference in the manuscript (doi: 10.1371/journal.pone.0034847)

Comment: L361: Need to provide here and name/brand of polymerase used

Authors' response: Thank you very much for your and constructive suggestions. We have added in the manuscript: The PCR amplification was performed using Taq DNA polymerase (Thermo Fisher Scientific, Cat. No. EP0402, 5 U/ μ L)

Comment: L366: replace "gelation recovery" for "extracted from the gel"

Authors' response: Thank you very much for your and constructive suggestions. We have revised this in the manuscript.

Comment: L366: refers to a single library being prepared and sequenced

Authors' response: Thank you very much for your and constructive suggestions. We have revised this in the manuscript. The purified PCR products of each sample were independently barcoded and then pooled together for sequencing library construction. The TruSeq™ DNA Sample Prep Kit was used to construct the sequencing libraries, ensuring that each of the 81 samples was individually labeled with a unique barcode. The pooled, barcoded samples were subsequently prepared and processed according to the manufacturer's protocol for high-throughput sequencing

Comment: L373: the word "denoised" does not exist, please use common

terminology for cleaning/curating sequences

Authors' response: Thank you very much for your and constructive suggestions. We replaced denoised with quality-filtered in the manuscript.

Comment: L378, L380, L382: Need reference for Shannon and Richness indices, for PCoA analysis, Wilcoxon test,

Authors' response: Thank you very much for your comments and suggestions. We have added the relevant references to the manuscript.

Comment: L391: replace "supplemented" for "supplemented with"

Authors' response: Thank you very much for your and constructive suggestions. We have revised this in the manuscript.

Comment: L392: the antibiotics are listed using different units (units, micrograms), please double check if this was intended

Authors' response: Thank you for your question. The use of this antibiotic is based on the article (doi: 10.1016/j.cell.2024.04.043). The use of different units is to facilitate better reproducibility of the experiment.

Comment: L396-397: needs more clarity detailing what was done

Authors' response: Thank you very much for your and constructive suggestions. We have added relevant details in the manuscript: To ensure that the isolated bacteria originate solely from the mouse intestine, the cecal contents were promptly diluted and inoculated into the culture medium. All experiments were conducted in a clean laboratory equipped with an independent filtration and ventilation system, and all sample collection consumables were disposable and sterile.

Comment: L404: please use capital letter for Martin

Authors' response: Thank you very much for your and constructive suggestions. We have revised this in the manuscript.

Comment: L406: please explain how the precipitates were obtained

Authors' response: Thank you very much for your comments, and we have added the method for obtaining the fungal precipitate in the manuscript: When the fungi grow to a suitable concentration (e.g., when a cloudy suspension of fungi can be observed in the liquid culture medium), take an aliquot of the culture medium and transfer it to a

centrifuge tube. Centrifuge at 3000-5000 x g for 10-15 minutes. After centrifugation, carefully pour off the supernatant to obtain the fungal precipitate.

Comment: L153, L416: replace "sanger" for "Sanger"

Authors' response: Thank you very much for your and constructive suggestions. We have revised this in the manuscript.

Comment: L419: isolates need to be placed in a public database to make them accessible to the scientific community, please provide Accession numbers

Authors' response: Thank you very much for your comments and suggestions. We have deposited the ITS1 sequences and annotation files of all isolates in the Zenodo repository with access number 14784111 (<https://doi.org/10.5281/zenodo.14784111>).

Comment: L447: It is possible that PCR could have generated hybrid amplicons? Please consider.

Authors' response: Thank you very much for your question. The primers we used in this study are highly specific to the ITS1 region and can effectively avoid nonspecific binding to other non-target DNA sequences, thereby reducing the generation of hybrid amplicons. In addition, we optimized the PCR conditions and used specific annealing temperatures and cycling conditions to improve the specificity of amplification. To ensure that only the target region is amplified. In addition, before Sanger sequencing, 2% agarose gel electrophoresis was used to verify the PCR products to ensure the size and specificity of the amplified products and further exclude the possibility of hybrid amplicons.

Despite the above measures, low-abundance chimeras may still remain in complex samples. Future studies can further combine long-read sequencing technology (such as PacBio HiFi or Oxford Nanopore) to directly obtain full-length amplicons to improve specificity.

We have corrected other minor mistakes and typos in the manuscript.

We sincerely hope that the manuscript has been revised to your satisfaction and we are looking forward to receiving your editorial decision.

Sincerely yours,

Xiao-Xuan Zhang

College of Veterinary Medicine,

Qingdao Agricultural University, Qingdao,

Shandong Province 266109, PR China

Email: zhangxiaoxuan1988@126.com

Re: Spectrum02840-24R1 (Investigation of the genetic diversity of gut mycobiota of the wild and laboratory mice)

Dear Prof. Xiao-Xuan Zhang:

The reviewers have read the modifications and have a minor comment. Could you please address this comment and return the manuscript for final decision. Below you will find my comments, instructions from the Spectrum editorial office, and the reviewer comments.

Please return the manuscript at your earliest convenience; if you cannot complete the modification within this time period, please contact me. If you do not wish to modify the manuscript and prefer to submit it to another journal, notify me immediately so that the manuscript may be formally withdrawn from consideration by Spectrum.

Revision Guidelines

Sincerely,
Renee Arias
Editor
Microbiology Spectrum

Reviewer #1 (Comments for the Author):

Response to the following comments is satisfactory. However, it was not incorporated into the manuscript. It is critical to the reproducibility of the work. Your response to my queries should be incorporated into the relevant sections of the manuscript.

Comment:

Line 350: Overall, please add additional metadata (age, length, weight, date of collection etc.) about mice if available. Also provide additional details of laboratory mice husbandry (food, housing conditions like temperature, humidity etc.).

Comment:

Line 456: Any trimming of the alignment was performed before phylogenetic analysis. How did you select the best for phylogenetic analysis and specify the models used? Also determine the number of bootstraps and how the tree was visualized (software?)

Reviewer #2 (Comments for the Author):The modifications have been addressed, though I am not sure if the assembled genomes were uploaded to the databases.

It is not clear whether the assembled genomes were included in the data uploaded to NCBI.

February 28, 2025

Dr. Renee Arias

Co-Editor-in-chief

Microbiology Spectrum

Re: Revised Manuscript ID Spectrum02840-24R2

Dear Dr. Renee Arias

On behalf of all authors, I would like to thank the Editor and two reviewers for their comments and constructive suggestions on our **manuscript ID Spectrum02840-24R1**. We have significantly revised the manuscript based on the reviewers' comments, corrected all errors, added experimental details, and supplemented the data. In the next section, we will detail our point-by-point responses to specific comments and suggestions.

Responses to the comments and suggestions of reviewer #1:

Reviewer #1 (Public repository details (Required)):

Comment: Response to the following comments is satisfactory. However, it was not incorporated into the manuscript. It is critical to the reproducibility of the work. Your response to my queries should be incorporated into the relevant sections of the manuscript.

Authors' response: Thank you very much for your valuable comments. We have added detailed responses to the following two questions in the manuscript.

Comment: Line 350: Overall, please add additional metadata (age, length, weight, date of collection etc.) about mice if available. Also provide additional details of laboratory mice husbandry (food, housing conditions like temperature, humidity etc.).

Authors' response: Thank you very much for your valuable comments. Regarding this comment, we have added the following content to the manuscript: All C57BL/6J (purchased from (Beijing) Biotechnology Co., Ltd.) mice were housed in a specific pathogen-free facility with controlled environmental temperature and humidity (22

$\pm 2^{\circ}$ C, 50% \pm 10%), a 12:12 h light/dark cycle, and an independent ventilation system, with food and water available ad libitum. Detailed information on the mouse species, sex, age, weight, sampling time, latitude and longitude of the sampling area, and altitude is shown in Supplementary Table 1.

Comment: Line 456: Any trimming of the alignment was performed before phylogenetic analysis. How did you select the best for phylogenetic analysis and specify the models used? Also determine the number of bootstraps and how the tree was visualized (software?)

Authors' response: Thank you very much for your valuable comments. Regarding this comment, we have added the following content to the manuscript: The protein markers of each genome are combined into concatenated sequences to construct a phylogenetic tree. The specific process of constructing a phylogenetic tree is as follows: 1) Multiple Sequence Alignment and Trimming: Alignment Tool: Sequences were aligned using MAFFT (v7.475) with the --auto parameter to automatically select the optimal alignment strategy. No additional trimming was performed, as IQ-TREE's model selection and tree inference account for variable sites during analysis. 2) Model Selection and Phylogenetic Tree Construction: Software: Phylogenetic analysis was conducted using IQ-TREE (v2.1.2). Model Selection: The best-fit substitution model was automatically selected by IQ-TREE's built-in ModelFinder algorithm, which evaluates models using the Bayesian Information Criterion (BIC). Bootstrap Analysis: Branch support was assessed using ultrafast bootstrap approximation (UFBoot) with 1000 replicates, a default setting in IQ-TREE that balances computational efficiency and robustness. 3) Tree Visualization: Visualization Tool: The final maximum-likelihood tree was visualized and annotated using iTOL (Interactive Tree of Life, v5).

Responses to the comments and suggestions of reviewer #2:

Reviewer #2 (Public repository details (Required)):

Comment: The modifications have been addressed, though I am not sure if the

assembled genomes were uploaded to the databases. It is not clear whether the assembled genomes were included in the data uploaded to NCBI.

Authors' response: Thank you for your question. In order to facilitate readers to obtain fungal genome assembly data more conveniently, we have uploaded 48 fungal genome assembly data to the Zenedo database for public access. Accession number: 14939226 (<https://doi.org/10.5281/zenodo.14939226>). We have added this accession number to the manuscript

We have corrected other minor mistakes and typos in the manuscript.

We sincerely hope that the manuscript has been revised to your satisfaction and we are looking forward to receiving your editorial decision.

Sincerely yours,

Xiao-Xuan Zhang

College of Veterinary Medicine,

Qingdao Agricultural University, Qingdao,

Shandong Province 266109, PR China

Email: zhangxiaoxuan1988@126.com

Re: Spectrum02840-24R2 (Investigation of the genetic diversity of gut mycobiota of the wild and laboratory mice)

Dear Prof. Xiao-Xuan Zhang:

Your manuscript has been accepted, and I am forwarding it to the ASM production staff for publication. Your paper will first be checked to make sure all elements meet the technical requirements. ASM staff will contact you if anything needs to be revised before copyediting and production can begin. Otherwise, you will be notified when your proofs are ready to be viewed.

Sincerely,
Renee Arias
Editor
Microbiology Spectrum